# Knowledge of Medical Imaging Professionals on Healthcare-Associated Infections: A Systematic Review and Meta-Analysis

**DOI:** 10.3390/ijerph20054326

**Published:** 2023-02-28

**Authors:** Suresh Sukumar, Shovan Saha, Winniecia Dkhar, Nitika C. Panakkal, Visakh Thrivikraman Nair, Tulasiram Bommasamudram, K Vaishali, Ravishankar Nagaraja, Sneha Ravichandran, Rajagopal Kadavigere

**Affiliations:** 1Department of Medical Imaging Technology, Manipal College of Health Professions (MCHP), Manipal Academy of Higher Education (MAHE), Manipal 576104, India; 2Department of Occupational Therapy, Manipal College of Health Professions (MCHP), Manipal Academy of Higher Education (MAHE), Manipal 576104, India; 3Department of Exercise and Sports Sciences, Manipal College of Health Professions (MCHP), Manipal Academy of Higher Education (MAHE), Manipal 576104, India; 4Department of Physiotherapy, Manipal College of Health Professions (MCHP), Manipal Academy of Higher Education (MAHE), Manipal 576104, India; 5Department of Biostatistics, Vallabhbhai Patel Chest Institute, University of Delhi, Delhi 110021, India; 6Department of Radiodiagnosis and Imaging, Kasturba Medical College (KMC), Manipal Academy of Higher Education (MAHE), Manipal 576104, India

**Keywords:** knowledge, attitude, healthcare-associated infection, medical imaging professionals, occupational health, risk assessment

## Abstract

Healthcare-associated infections (HCAIs) are a significant concern for both healthcare professionals and patients. With recent advances in imaging modalities, there is an increase in patients visiting the radiology department for diagnosis and therapeutic examination. The equipment used for the investigator is contaminated, which may result in HCAIs to the patients and healthcare professionals. Medical imaging professionals (MIPs) should have adequate knowledge to overcome the spread of infection in the radiology department. This systematic review aimed to examine the literature on the knowledge and precaution standard of MIPs on HCIAs. This study was performed with a relative keyword using PRISMA guidelines. The articles were retrieved from 2000 to 2022 using Scopus, PubMed, and ProQuest databases. The NICE public health guidance manual was used to assess the quality of the full-length article. The search yielded 262 articles, of which Scopus published 13 articles, PubMed published 179 articles, and ProQuest published 55 articles. In the present review, out of 262 articles, only 5 fulfilled the criteria that reported MIPs’ knowledge of Jordan, Egypt, Sri Lanka, France, and Malawi populations. The present review reported that MIPs have moderate knowledge and precautionary standards regarding HCIAs in the radiology department. However, due to the limited studies published in the literature, the current review limits the application of the outcome in the vast MIPs population. This review recommended further studies to be conducted among the MIPs worldwide to know the actual knowledge and precaution standards regarding HCIAs.

## 1. Introduction

Medical imaging professionals, such as radiologists, radiologic technologists, and other imaging specialists, may have some knowledge of healthcare-associated infections (HAIs). However, their level of knowledge may vary depending on their specific area of expertise and their exposure to patients with HAIs. HAIs are infections that patients acquire while receiving medical care, and they can occur in any healthcare setting, including hospitals, outpatient clinics, and long-term care facilities. HAIs are a significant public health concern, and healthcare professionals across all specialties are responsible for preventing and controlling these infections. Medical imaging professionals may have some knowledge of HAIs because they work closely with patients and often come into contact with contaminated equipment and surfaces. They may be trained in infection control measures, such as proper hand hygiene, cleaning and disinfection of equipment, and the use of personal protective equipment (PPE) to reduce the risk of transmission. In addition, medical imaging professionals may be involved in the diagnosis and treatment of patients with HAIs, particularly those that affect the respiratory system, such as tuberculosis or pneumonia. They may use imaging techniques such as chest X-rays, computed tomography (CT), or magnetic resonance imaging (MRI) to detect and monitor the progression of these infections. Overall, medical imaging professionals are an essential part of the healthcare team and play a critical role in preventing and controlling HAIs. While their specific knowledge of HAIs may vary, they should receive ongoing education and training on infection control measures to ensure the safety of both patients and healthcare workers.

The pandemic outbreak of Zoonotic viral diseases such as Corona, Nipah, and Ebola has resulted in a public health emergency with months of lockdown and financial loss to major tax-collecting departments such as tourism, aviation, and import–export. Further, the Zoonotic viral diseases have threatened the world economy with the loss of manpower, funds rerouting to the healthcare sector and vaccine research, and the health of human beings, leading to mass deaths and hospitalizations [1]

The World Health Organization (WHO) notified COVID-19 as a pandemic disease on 11 March 2020, and the global containment and quarantine efforts contamination incidents continue to increase post-haste. In developing countries such as India, there is a noticeable increase in the in-flow of infected patients to the hospital due to the recent and past outbreaks of pandemic Zoonotic viral diseases. The chances of cross infections increase with more immunocompromised patients visiting the hospital for other underlying cases. Still, they become infected with nosocomial infections, creating the need to assess cross-infection locations.

The infected patients with Zoonotic viral diseases primarily presented with dyspnoea, fever, and dry cough [2]. Further, these patients are highly contagious, and the infection is transmitted between people through droplets, close contact, and fomite [3,4]. The virus-like COVID-19 remains viable for 72 h on plastic and stainless steel and up to 3 h in aerosols [5]. The estimated half-life of the virus in the aerosol ranged from 1.1 to 2.1 h. Metals such as stainless steel have a half-life of 5.5 h, and the half-life of the plastic is 6.8 h [6]. The presence of bioaerosol in the surrounding air has profoundly influenced the health of humans, animals, and plant life. Further, both the viable and nonviable pathogens such as bacteria, fungi, and viruses present in the surrounding environmental air may also influence health [7,8].

Colonized and highly contagious infected patients visiting the hospital for different treatments may increase the chance of HCAI for patients and healthcare workers. Developing countries have a much higher risk of HCAI, with a radio of 20:1 as compared to developed countries [9]. In the past, HCAI was limited to inpatients. However, the recent pandemic virus outbreak resulted in increased HCAI in outpatients. Since the Department of Radiodiagnosis and Imaging plays a significant role in diagnosing different infectious diseases, there is a high probability of HCAI among radiology staff and patients [10]. Equipment such as radiography, magnetic resonance imaging, computed tomography, fluoroscopy, ultrasound, echocardiography and positron emission tomography used for diagnostic and therapeutic examinations were more prone to infection, resulting in healthcare-associated infections (HCAI). The other infected equipment in the Radiology Department (RD) including the imaging tables, keyboard, touchscreen, computer mouse, lead apron, radiographic markers, and transferring table are also more prone to the HCAIs [11,12,13,14,15,16,17,18].

However, despite the improvement in the practices, HCAI is still prevalent in the healthcare facilities affecting patients worldwide each year. In the recent outbreak, HCAI has reported that 3.8% of the total infections are related to COVID-19 [19]. To overcome the challenges of HCAIs, the knowledge and precautionary standards regarding HCPs must improve [20]. The HCPs such as Medical Imaging Professionals (MIP) working in the radiology equipment should know standardized operating protocols to minimize the spread of HCAIs via the radiology equipment [21,22].

Since there is limited research regarding the knowledge of the HCAI among the HCPs reported in the literature, the awareness of MIPs on the HCAI has also been underexplored.

The purpose of this study is to interpret research reports pertaining to knowledge and precaution standards among MIPs.

## 2. Materials and Methods

This systematic review was performed to identify MIPs’ knowledge and precautionary standards regarding HCAIs in the radiology department. PRISMA-reporting guidelines were followed in this study [23].

### 2.1. Search Strategy

The electronic databases PubMed, Scopus, and ProQuest were considered to retrieve the studies reported from 2000 to 2022. The literature search was performed from 1 June 2022 to 30 June 2022 and restricted to studies published in English. The keywords and search strategy used are presented in Table 1. A manual search was also performed to identify the relevant articles using the included studies’ citations.

### 2.2. Selection Criteria

Research papers focused on the “knowledge”, or “education”, or “understanding”, or “awareness” of the HCAI among the MIPs were included. Articles published in the English language were considered for this study. The full-length research papers were independently reviewed by the two reviewers and then shortlisted. The disagreement between the reviewers over eligibility were resolved through discussion and concession with the third reviewer.

#### 2.2.1. Selection of the Studies

The title and abstract of the articles were appraised for relevance to the aim of the present review. The articles that were not relevant were eliminated. The full text of the remaining article fulfilled the objective of the evaluation and was acquired to assess further relevance based on the inclusion and exclusion criteria. The full-length research papers were independently reviewed by the two reviewers and then shortlisted. The third reviewer verified shortlisted articles based on the review criteria. The disagreement between the reviewers over eligibility was resolved through discussion and concession with the third reviewer.

#### 2.2.2. Data Extraction

The data of the included full-length articles were extracted into the Microsoft Excel sheet. The year, country, samples, study design, number of items present, validation of the questionnaires, ethical approval of the studies, and mode of data collection were considered during the data extraction.

#### 2.2.3. Risk of Bias

The NICE public health guidance manual was used to assess the risk of bias for the included studies [21,22] (Table 2). Two reviewers performed the assessments independently, and the correlation was determined.

#### 2.2.4. Criteria for Inclusion and Exclusion

Exclusion: focus on HCP, not focus on medical imaging, other language.

Inclusion: focus on the medical imaging, published in English.

#### 2.2.5. Data Analysis

The Kappa test was performed to determine the two reviewers’ agreement using SPSS 16. A meta-analysis was used to obtain the pooled estimate. The random-effects model was used for the meta-analysis. Chi-square statistics and I-square statistics were used to report heterogeneity. A meta-analysis was performed using STATA software version 13.1.

## 3. Results

The magnitude of the publication related to the knowledge and standard precautions of HCAIs among the MIPs was initially unknown. Therefore, the present research question was focused on MIPS. The literature searches across the three electronic databases resulted in 247 articles in the current review. The additional 14 articles were obtained by the manual search based on the references. Once the duplicate papers were eliminated, the study number was reduced to 238 articles. A total of 221 articles that were not focused on the MIPs were removed. The full text of the remaining 11 articles [24,25,26,27,28,29,30,31,32,33,34] was collected and screened for the inclusion and exclusion criteria, out of which only 5 articles are eligible for review and evaluation [24,25,29,30,32]. The PRISMA is presented in Figure 1 [23].

### 3.1. Characteristics of the Included Studies

A total of five articles published from Jordan, Egypt, Sri Lanka, France, and Malawi were considered for the present review [24,25,29,30,32]. All the articles were published between 2008 and 2018 (Table 3). Out of five studies, only two studies reported the questionnaire specific to the MIPs’ knowledge of infection control practices specific to the RD. However, all the items included in the questionnaire were validated before the data collection. The questionnaire used in the studies to evaluate the knowledge and precautionary standards varied from 18 to 50 items. All the included studies reported that ethical approval was obtained to collect the data via a self-administered survey.

### 3.2. Methodological Quality

The quality of the selected articles has been summarized in Table 4. All five studies were found to be of quality ++. A moderate agreement was reported between the two reviewers, with a kappa value of 0.545.

### 3.3. Meta-Analysis

In the present review, out of five included studies, only three studies [25,29,32] were considered for the meta-analysis. The pooled effect of the outcome measures related to the “Environment as a major source of infection,” “Invasive procedures increase the risk of nosocomial infection,” “Gloves recommendation” and “Precaution standards during the procedure” is reported in Table 5. The studies which did not report the specific outcome measures were excluded from the meta-analysis [24,30].

### 3.4. Environment and HCIAs

The MIPs show poor knowledge with a pooled effect size of 0.18 (0.08, 0.28). Considerable heterogeneity is present among the studies, with I^2^ of 85.39% and *p* ≤ 0.001 (Figure 2).

### 3.5. Gloves Recommendation during the Procedure

The MIPs reported poor knowledge with a pooled effect size of 0.41 (0.17, 0.65). Considerable heterogeneity is present among the studies included, with I^2^ of 96.25% and *p* ≤ 0.001 (Figure 3).

### 3.6. Precaution Standards

The MIPs possess excellent knowledge with a pooled effect size of 0.87 (0.77, 0.97). Considerable heterogeneity is present among the studies included, with I^2^ of 96.96% and *p* ≤ 0.001 (Figure 4).

In the present review, the medical imaging professional shows good knowledge, with a pooled effect size of 0.76 (0.44, 1.07). Considerable heterogeneity is present among the studies included, with I^2^ of 98.91% and *p* ≤ 0.001 (Figure 5).

## 4. Discussion

This systematic review was aimed at investigating the knowledge of the MIPs on the HCAIs. The literature search resulted in five articles, of which three articles were considered for the meta-analysis.

### 4.1. The Spread of Infection

The RD plays a vital role in the diagnosis and treating of the patient’s disease using X-ray, CT scanner, MRI Scanner, interventional radiology, and ultrasound (USG). The probability of the HCAIs may increase due to the colonized and infected patients waiting for the diagnostic procedure [13]. Further, the contaminated hand is considered to be one of the primary sources of transferring the HCAIs. The contaminated hand of the MIPs and the other HCPs, including nurses, practitioners, and attendees, can lead to the transfer of the infection pathogens from one person to another person, adjacent surface, and equipment [35]. Nyirenda et al. reported that 98% of Malawi MIPs have good knowledge about the primary source and spread of infection [30]. However, in the present review, the MIPs reported that the environment was a major source of multi-resistant bacterial transmission, with a pooled effect size of 0.18 (0.08, 0.28). Giacometti et al. reported that 41.7% of the X-ray tube, 91.7% of control panels and imaging plates, and 8% of the X-ray cassettes were contaminated, which may transmit the HCAIs [36]. Therefore, faculty working in the RD should have updated knowledge and standard operating procedure to reduce the HCAIs spread via radiology equipment or the HCP in the RD.

### 4.2. Knowledge of Hand and Machine Hygiene in RD

Olu O. et al. [37] reported 95% of healthcare professionals acquired the HCAI during the outbreak of Ebola in 2014. Therefore, the HCAI in the Healthcare Professional (HCP) may accelerate the transmission and result in mortality [5]. However, if healthcare professionals adhere to the guidelines and protocol of disinfection and isolation, the risk of HCAI may drop to 10.7% [37]. The HCAI can be overcome by hand hygiene and sanitation practices, which may lower the incidence of the disease, as opposed to treating the disease [38].

The microbial growth on an MIP’s hand can be effectively overcome by sterilization of the hand. The alcohol-based hand sanitisers were most commonly used for the sterilization of hand fields [13]. Further, proper sterilization of the imaging equipment can prevent one-third of the HCIAs [36]. Carling et al. reported that 88% of MIPs were aware that the primary reason for the contamination of the radiology equipment is due to the inadequate disinfection procedures followed in the department [39]. Cleaning the equipment and cassette between the patient’s examinations using the alcohol wipe and chlorhexidine-based detergents is effective, and they are recommended as compared to soap and water [40,41]. The alcohol gel, disinfecting wipes, and the standard hand wash are recommended to disinfect the radiographer’s markers during the procedure [40]. However, special consideration is required for the ribbon markers [42].

The American Institute of USG in medicine recommended using water and soap or quaternary ammonium disinfectant spray for the USG [40]. The MRI machine’s disinfection is difficult compared to the other imaging equipment in the RD [41]. However, it is recommended to use the solution of 1000 parts of hypochlorite with a million parts of chloride to disinfect the MRI machine [11]. Abdel-Hady El-Gilany et al. reported that the MIPs have inadequate knowledge about sanitation practices as compared to the other HCPs [24].

During the portable examination of the infected patients, especially EBOLA and COVID-19 patients, specific emphasis on hand hygiene is necessary [35]. Further, additional precaution measures include the proper use of gloves while performing and handling high-risk patients. The gloves are also recommended when there is a risk of MIPs in contact with blood during the procedure [11]. However, it is not recommended to use gloves for every radiological procedure [25]. In the present review, the MIPs show poor knowledge of the glove’s recommendation, with a pooled effect size of 0.41 (0.17, 0.65).

### 4.3. Recommendations of Precaution Standards to Protect the Patients and HCP

The MIPs can break the chain of HCAIs from other HCPs through proper education, hand hygiene surveillance, vaccination against preventable disease, and prevention of needle-struck injury. Abdel-Hady El-Gilany et al. reported that the MIPs have inadequate knowledge about needle-struck injury as compared to the other HCPs [24].

Training the MIPs with high-quality education can break the spread of the HCAIs in the radiology department [22]. Bello et al. reported that only half of the MIPs have intermediate knowledge of infection control measures [43]. However, in the present study, the MIPs show poor knowledge about the “Precaution standards recommendations to protect the patients and the healthcare workers” with a pooled effect size of 0.87 (0.77, 0.97).

### 4.4. Invasive Procedures and HCIAs

Interventional radiology is considered a high-risk area for imparting HCAIs [36,44]. The radiation protective equipment used during the intervention procedure has also been observed to be more prone to HCAIs [45]. The reduction in infection in interventional radiology requires the identification of the risk, the appropriate use of antibiotic prophylaxis, and appropriate patient care [46]. In the present review, the medical imaging professional shows good knowledge about invasive procedures that increase the risk of HCIAs, with a pooled effect size of 0.76 (0.44, 1.07).

Overall, the discrepancy in the result is present among the studies included in the current review. The MIPs have poor knowledge of the spread of HCIAs infection and the gloves recommendation. However, MIPs reported good knowledge of precaution standards to protect the patients and healthcare workers.

### 4.5. Limitations of the Included Studies

According to the hierarchy of evidence, the cross-section and observational study design ranked low as compared to the randomized control trials [42]. All five studies included in this review are of the cross-section study design that typically lacks the adequate methodology rigour that minimizes the effect of bias. However, applying a rigid design hierarchy to this research is potentially less significant than evaluating the precision of the study methodology [47]. Of the five systematic review studies, one used a sample of fewer than fifty subjects. However, no studies justified the sample size. The small sample size may result in the probability of non-significant results [48]. Out of five studies, only one was reported using random sampling techniques. This further limits the further the ability to find the wider population. Of the rest of the studies, one used convenience sampling, and four did not state the sampling procedure. The ability to extrapolate the survey to the other MIPs is difficult due to the limited number of studies. Variable types of items in the questionnaire were used in the surveys. In the current review, three out of five studies used similar questions, of which only one study reported specific to the MIPs.

### 4.6. Limitation of the Present Review

In the present review, the literature search was performed based on only three electronic databases (PubMed, Scopus, and ProQuest). Further, the systematic review was limited to the articles published between 2000 to 2022 and restricted to studies published in English. Due to the limited number of papers published and the diversity in the items present in the questionnaire, a meta-analysis for the present study was performed with three articles with similar items. However, the outcome of the current review on MIPs’ knowledge and precautionary standards regarding HCAIs may limit the application of the result to the vast population.

### 4.7. Future Recommendations from the Present Review

Based on the current review, several research gaps have been identified. A minimal study was undertaken and published worldwide. Further research can be performed to develop a valid tool for understanding the MIPs’ knowledge specific to the radiology department in preventing the HCAIs. All the studies included in this review are questionnaire-based quantitative studies. There was no study reported in the qualitative research methodology. Conducting the focus group discussion may result in a better understanding of the knowledge of MIPs on HCAIs. The present review has not focused on other allied health professionals. A systematic review needs to be undertaken among all the allied health professionals. Further, a study on infection control with proper guidelines and the HCP’s compliance in involving these measures may be imperative in reducing the HCAIs, thereby reducing the burden and improving both HCP and patient outcomes [15].

### 4.8. Future Scopes

The qualitative and quantitative study on the awareness of the spread of infection among MIPs may provide better knowledge and understanding among imaging professionals. Diversity in the items in the questionnaire was observed; therefore, a standardized questionnaire must be developed to assess knowledge, awareness, and practice of handling infectious diseases in the Radiology Department. Further reviews have to be conducted on MIPs’ understanding of the HCAI and the application of modern imaging techniques during the outbreak of infectious disease to reduce the risk of HCAI to MIPs.

## 5. Conclusions

The current review aimed to understand MIPs’ knowledge and the precaution standards regarding the HCAIs in the RD. The present study resulted in five articles which fulfilled the inclusion criteria. Of these, only three used similar items related to the spread of infection, knowledge of precaution standards and hygiene, and risk of disease in invasive procedures. The present study reported that MIPs have moderate knowledge and precautionary standards toward the HCIAs in the RD. However, due to the limited studies published in the literature, the outcome of the current review may limit the application of the outcome results in the vast MIPs population worldwide. This review recommended further qualitative and quantitative studies to be conducted among imaging professionals worldwide to discern the actual knowledge and precaution standards regarding HCIAs.

## Figures and Tables

**Figure 1 ijerph-20-04326-f001:**
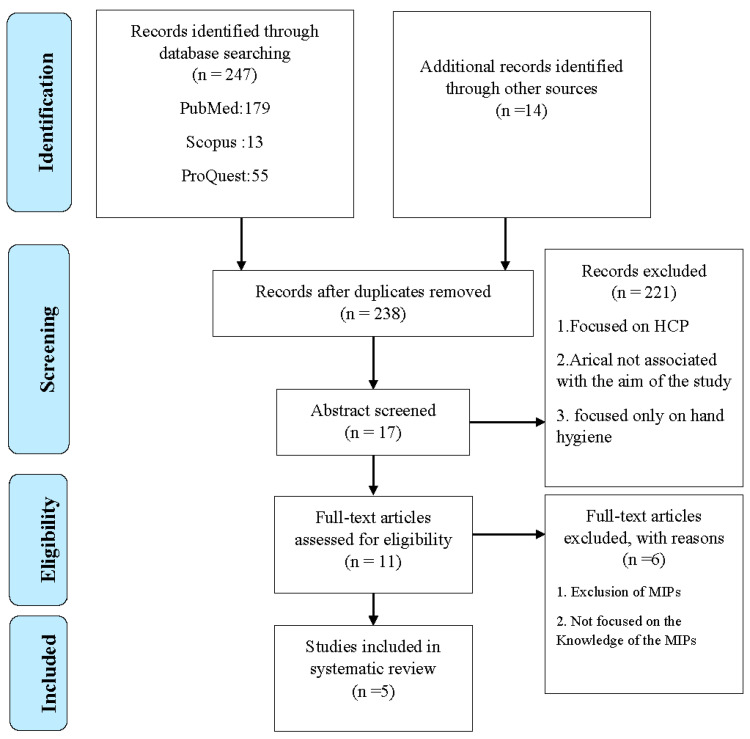
Flow diagram of the article selection process.

**Figure 2 ijerph-20-04326-f002:**
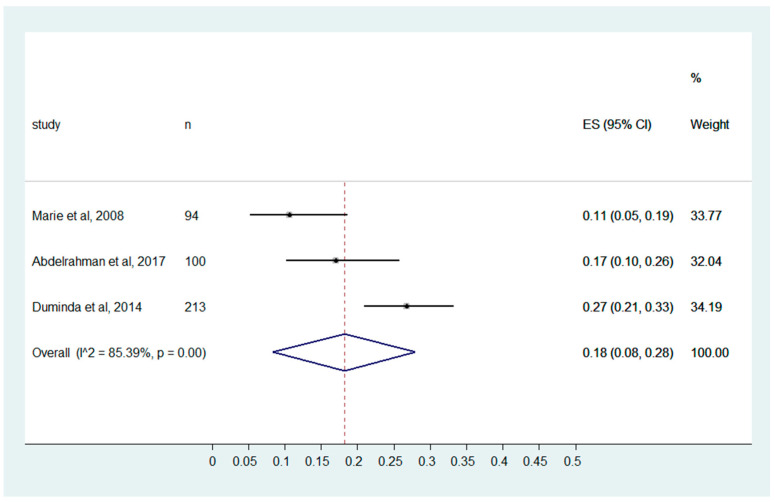
MIPs’ knowledge of “The environment (air, water, inert surfaces) is the major source of bacteria responsible for nosocomial infection”. (Marie et al., 2008 [32], Abdelrahman et al., 2017 [25], Duminda et al., 2014 [29]).

**Figure 3 ijerph-20-04326-f003:**
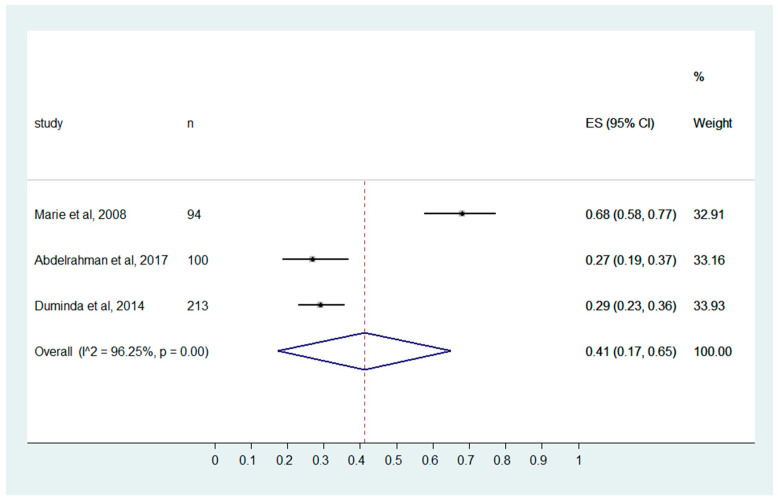
MIPs knowledge on “Gloves recommendation during the procedure.” (Marie et al., 2008 [32], Abdelrahman et al., 2017 [25], Duminda et al., 2014 [29]).

**Figure 4 ijerph-20-04326-f004:**
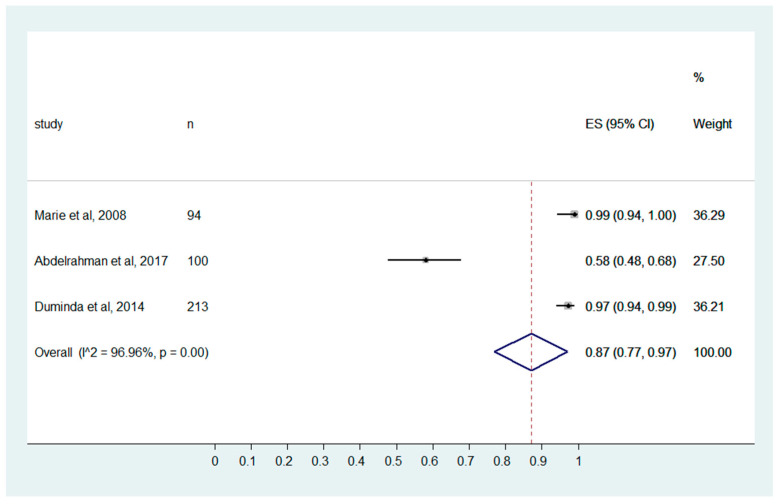
MIPs’ knowledge of “Precaution standards to protect the patients and the HCPs”. (Marie et al., 2008 [32], Abdelrahman et al., 2017 [25], Duminda et al., 2014 [29]).

**Figure 5 ijerph-20-04326-f005:**
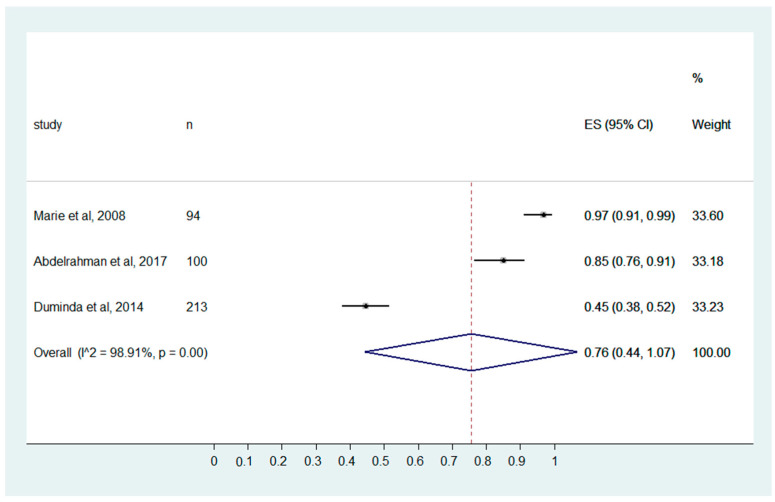
MIPs’ knowledge of “Invasive procedures increase the risk of HCAIs”. (Marie et al., 2008 [32], Abdelrahman et al., 2017 [25], Duminda et al., 2014 [29]).

**Table 1 ijerph-20-04326-t001:** Keywords Strategy.

Sl. No.	Search
1	“(nosocomial infections or hospital-acquired infections or healthcare-associated infections)”
2	“(knowledge or education or understanding or awareness)”
3	“ (radiographer or radiologist technologist).”
	1 AND 3
	1 AND 2 AND 3
4	“ (nosocomial Infections)”
5	“(knowledge) “
6	“ (radiographer)”
	4 AND 5 AND 6

**Table 2 ijerph-20-04326-t002:** Criteria to assess the risk of bias.

Criteria’s	Score
Clarity of the questioners	+/−
Details of the study methodology	+/−
Details of data collection	+/−
Research content	+/−
Data analysis	+/−
Result revenant to the study’s objective	+/−
Ethics approval	+/−
Overall	++/+/−

++ must meet at least six criteria indicated above. + must meet at least four criteria mentioned above. ++ denotes the low level of bias; + means the moderate level of bias; and − indicates a poor level of bias.

**Table 3 ijerph-20-04326-t003:** Characteristics of the included studies.

	Author				
	M.A. Abdelrahman et al. [25]	El-Gilany et al. [24]	Jayasinghe et al. [29]	Marie-Pierre et al. [32]	Nyirenda, D. et al. [30]
Year	2017	2012	2014	2008	2018
Country	Jordan	Egypt	Sri Lanka	France	Malawi
Samples	128	14	213	94	62
Study design	CS	CS	CS	CS	CS
No of items	41	50	18	25	36
Ethical approval	Yes	Yes	Yes	Yes	Yes
Mode of data collection	SA	SA	SA	SA	SA
Validation	Yes	Yes	Yes	Yes	Yes
Items specific to radiology	Yes	No	No	No	No
Environment and HCAIs	Poor	*	Poor	poor	Good
Intervention and HCAIs	Good	*	Moderate	Good	*
Standard precaution and HCAIs	Good	Moderate	Good	Good	Moderate
Gloves recommend and HCAIs	Poor	*	Poor	Moderate	*

* indicates Not reported, and CS indicates a Cross-sectional study, SA—Self Administered.

**Table 4 ijerph-20-04326-t004:** Quality of the articles.

Study	Items							
	1	2	3	4	5	6	7	Overall
M.A. Abdelrahman et al. [25]	+	+	+	+	+	+	+	++
El-Gilany et al. [24]	+	+	+	+	+	+	+	++
Jayasingheet al. [29]	+	+	+	+	-	+	+	++
Marie-Pierre et al. [32]	+	+	+	+	+	+	+	++
Nyirenda, D. et al. [30]	+	+	+	+	-	+	+	++

1—Clarity of the questioners, 2—Details of the study methodology, 3—Details of the data collection, 4—Research content, 5—Data analysis, 6—Result relevant to the study’s objective, 7—Ethics approval. ++ denotes the low level of bias; + means the moderate level of bias; and − indicates a poor level of bias.

**Table 5 ijerph-20-04326-t005:** The effect size of the outcome measures.

Outcome Measure	Effect Size (95% CI)
“The environment (air, water, inert surfaces) is the major source of bacteria responsible for nosocomial infection.”	0.18 (0.08, 0.28)
“Invasive procedures increase the risk of nosocomial infection.”	0.76 (0.44, 1.07)
“Medical Imaging Professionals knowledge on standard precautions recommend the use of gloves.”	0.41 (0.17, 0.65)
“Precaution standards to protect the patients and the healthcare workers.”	0.87 (0.77, 0.97)

## Data Availability

The data for this review is available in “Sukumar, Suresh, 2022, Knowledge of Medical Imaging Professionals on Healthcare-Associated Infections: A systematic review and meta-analysis, https://doi.org/10.7910/DVN/OTG4YZ”.

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
