# Peer review of "Knowledge of Medical Imaging Professionals on Healthcare-Associated Infections: A Systematic Review and Meta-Analysis"

_ijerph, 2023, doi:10.3390/ijerph20054326_

Round 1

Reviewer 1 Report

Thanks for submitting your manuscript. It is good to see an effort to explore the hospital acquired infections among the health care professionals. 

Introduction: after reading the introduction, I can see that you have most of the examples COVID-19 but the focus of the review is to know the knowledge of the medical imaging professionals. It would be good to focus on the medical imaging professionals mainly , in addition generally discuss the infection related issues among the health care professionals. No need to bring so many COVID-19 related information.

Methods: which framework was followed? is it Cochrane?

How the search terms are developed? What did you do with duplicate articles?

Why NICE guidelines was used?  You used meta analysis and used NICE guidelines? This was not clear to me when there are many strong risk assessment frameworks are available?

PRISMA will be part of the methods, not results.

Where is the data extraction table?

Author Response

Details of the suggested changes  

Title

Knowledge of Medical Imaging Professionals on Healthcare-Associated Infections: A systematic review and meta-analysis

Reviewer

SL.NO

Suggestions received

Clarifications

Line number

1

1

Introduction: after reading the introduction, I can see that you have most of the examples COVID-19 but the focus of the review is to know the knowledge of the medical imaging professionals. It would be good to focus on the medical imaging professionals mainly , in addition generally discuss the infection related issues among the health care professionals. No need to bring so many COVID-19 related information.

The  change has been incorporated

51-71

2

Methods: which framework was followed? is it Cochrane?

It is not Cochrane guidelines

3

How the search terms are developed? What did you do with duplicate articles?

  The duplicated articles were deleted.

4

Why NICE guidelines was used?  You used meta analysis and used NICE guidelines? This was not clear to me when there are many strong risk assessment frameworks are available?

The checklist used in this based on the NICE guidelines as mentioned in the NICE website. The detail guidelines is mentioned in the link below.

https://www.nice.org.uk/process/pmg6/resources/the-guidelines-manual-appendices-bi-2549703709/chapter/appendix-b-methodology-checklist-systematic-reviews-and-meta-analyses

5

PRISMA will be part of the methods, not results.

The change has been incorporated

Page no. 5

6

Where is the data extraction table?

A separate excel file has been attached below

author    Marie et al M.A. Abdelrahman et al Ruwan Duminda
year    2008 2017 2014
country    France Jordan srilanka
sample    94 100 213
question correction   Percentage of students who gave correct answer  
outcome 1        
Nosocomial infection -         
The environment (air, water, inert surfaces) is the major source of bacteria responsible for nosocomial infection. no  10.6 17 26.76
Advanced age or very young age increases the risk of nosocomial infection. yes 36.2 89 *
Invasive procedures increase the risk of nosocomial infection. yes  96.8 85 44.6
outcome 2        
Precaution standards        
Include the recommendations to protect only the patients no  97.8 81 *
Include the recommendations to protect the patients and the healthcare workers. yes 98.9 58 97.2
Apply for all the patients yes 75.5 82 *
outcome 3        
When is hand hygiene recommended? Outcome 2        
Before or after a contact with (or care of) a patient. no  98.9 * *
Before and after a contact with (or care of) a patient. yes 100 * *
Before  contact with (or care of) a patient. yes * 87 46.5
 After a contact with (or care of) a patient. yes * 98 51.2
Between patient contacts. yes 86.2 94 *
After the removal of gloves yes 75.5 84 *
outcome 4        
The standard precautions recommend use of gloves: coutcome 3        
For each procedure. no  68.1 27 29.1
When there is a risk of contact with the blood or body fluid yes 84 97 *
When there is a risk of a cut. yes 59.6 94 *
When healthcare workers have a cutaneous lesion. yes 67 91 *
outcome 5        
When there is a risk of splashes or spray of blood and body fluids, the healthcare workers must wear        
Only mask. no  9.7 78 *
Only eye protection no  8.6 55 *
Only a gown. no  1.1 63 *
Mask, goggles, and gown. yes 89.2 * 79.8
* not reported         

Reviewer 2 Report

The manuscript entitled as “Knowledge of Medical Imaging Professionals on Healthcare-Associated Infections: A systematic review and meta-analysis” is a systematic review that addresses a relevant topic about cross infections in departments of Radiology and Imaging. The authors followed PRISMA guidelines and performed a proper systematic review. However, this reviewer has major concerns regarding the results section. Bellow, the authors can find some suggestions and questions.

Introduction:

In the examples of zoonotic viral diseases, I suggest the addition of influenza viruses, since is one of the most frequent of the kind, responsible for thousands of deaths yearly.

Only a primary outcome was set: knowledge and precaution standards in preventing the HCAIs among the MIPs. In the reviewer’s opinion, objectives should be more specific and can be divided in primary and secondary outcomes.

Did the authors register this systematic review in PROSPERO database?

Methods:

Table’s legend should be above the table

Language should not be exclusion criteria

Includes the number of reviewers and solution for disagreement and more than two database, mesh terms included. The term “prevention” could be useful.

Specify inclusion and exclusion criteria in the methods section

Justify the choice of the random-effects model (in the results section, it is obvious by the statistics, however, it should be mentioned the reason of the choice).

If the heterogeneity was high, would make sense to divide the samples in subgroups (water, air) to lower heterogeneity and apply the fixed model?

Results:

Presentation of the results (line 168-190) can and should be improved. In some phrases, is written abbreviation forms and others, extended forms, confusing the reader. Several sub-divisions, with feel content within. I suggest a table or an alternative way to present the results. Distribute the images throughout the text.

As written in the introduction, only a primary outcome was set. However, in the results section, three different factors were analyzed and a meta-analysis of each one was performed. In addition, four graphics can be observed in the results, the environment, Gloves recommendation, Precaution standards and Invasive procedures. I suggest to the authors pointing out a primary outcome and some others as secondary. The backbone of a systematic review is a very clear and objective question and when it is not clearly set, readers might struggle to understand the obtained results.  

SA meaning in table 3 is lacking

Numbered items in table 4 should be detailed.

Discussion:

Duminda et al. study have more than double the number of patients in comparison to the other two studies. It can add bias in the meta-analysis result, especially with such a low number of articles included. Authors should disclose this finding.

Aleatory bold words throughout the text.

The division of the discussion is adequate; the authors propose some answers to the identified problems and limitations. I suggest adding some hypotheses to the causes as well; the reasons for this lack of knowledge or even a negligent behavior from healthcare professionals. 

Author Response

2

1

Introduction:

In the examples of zoonotic viral diseases, I suggest the addition of influenza viruses, since is one of the most frequent of the kind, responsible for thousands of deaths yearly.

 The  change has been incorporated

51-71

2

Only a primary outcome was set: knowledge and precaution standards in preventing the HCAIs among the MIPs. In the reviewer’s opinion, objectives should be more specific and can be divided in primary and secondary outcomes.

The  change has been incorporated

51-71

3

Did the authors register this systematic review in PROSPERO database?

NO

4

Table’s legend should be above the table

The suggestion has been incorporated

Table 1-5

5

Language should not be exclusion criteria

But for this study we included only English based articles since we are not able to read foreign languages like French, Spanish, or any other language

6

Includes the number of reviewers and solution for disagreement and more than two database, mesh terms included. The term “prevention” could be useful.

We accept your suggestion but if we include this new term then we have to do the whole review again. The whole table and the process has to repeated. But we assure that we will incorporate this term in out next review

7

Specify inclusion and exclusion criteria in the methods section

The  change has been incorporated

 180-182

8

Justify the choice of the random-effects model (in the results section, it is obvious by the statistics, however, it should be mentioned the reason of the choice)

The choice of a random-effects model over a fixed-effects model is typically made based on the assumption that the study participants or subjects included in the analysis represent a random sample from a larger population, and that there is some degree of heterogeneity or variability in the effect sizes across the studies included in the meta-analysis. A random-effects model allows for this heterogeneity by assuming that each study has its own true effect size, drawn from a distribution of effect sizes that reflects the overall variability in the population. This variability is then estimated and incorporated into the meta-analysis model, resulting in more conservative estimates of the overall effect size, wider confidence intervals, and greater statistical power. In contrast, a fixed-effects model assumes that the true effect size is the same across all studies, and that any observed variability is due to chance or sampling error. This assumption may not hold true in many cases, and can lead to biased estimates of the overall effect size if there is substantial heterogeneity across studies. Therefore, the choice of a random-effects model is often justified when there is a priori reason to believe that the effect sizes across studies are likely to be heterogeneous, or when statistical tests indicate significant heterogeneity. It is important to note, however, that the choice of model should always be based on careful consideration of the underlying assumptions and the characteristics of the data being analyzed.

9

If the heterogeneity was high, would make sense to divide the samples in subgroups (water, air) to lower heterogeneity and apply the fixed model?

The suggestion is accepted but to do any more changes in the division of the samples, the whole review has to be done again. So we will incorporate the suggestion In the future reviews

10

SA meaning in table 3 is lacking

SA means self administered

236

11

Numbered items in table 4 should be detailed.

The correction has been done

238-239

12

Discussion:

Duminda et al. study have more than double the number of patients in comparison to the other two studies. It can add bias in the meta-analysis result, especially with such a low number of articles included. Authors should disclose this finding.

We had to use this study in our meta analysis since we had only 5 articles at the end of the articles screening and removing this articles just for the reason of having more sample size will keep us inconclusive about the study and leave us with lack of studies  to perform the meta-analysis

13

Aleatory bold words throughout the text

The change has been incorporated

14

he division of the discussion is adequate; the authors propose some answers to the identified problems and limitations. I suggest adding some hypotheses to the causes as well; the reasons for this lack of knowledge or even a negligent behavior from healthcare professionals. 

The reasons and answers to this have been given in the discussion but it has been fragmented in the various paragraphs of the discussions

Round 2

Reviewer 1 Report

Thanks for making the changes. I am happy to accept the article.

Author Response

Thank you for your kind help

Reviewer 2 Report

The second version of the manuscript entitled as “Knowledge of Medical Imaging Professionals on Healthcare-Associated Infections: A systematic review and meta-analysis” brings some improvements and the incorporation of some suggestions. However, as a reviewer, I have to point out two concerns (n° 2 and 5) that are still to be addressed. Below, the authors can find my replies following the numbers of the authors’ response.

1 – I did not find influenza viruses within line 51-71 as pointed out by the authors. Nevertheless, it is just a suggestion.

2 – I apologize if the comment was not clear. For the reviewer’s point of view, the aims (line 119 – 122) are too broad for a systematic review. The authors should point out, as the aim of the study, the expected outcomes (e.g., what knowledge and precaution standards were researched), that is, knowledge of the environment, Gloves recommendation, knowledge of Precaution standards and knowledge of Invasive procedures.

3 – Ok. Although it is not mandatory to register, the authors would have some advantages in doing it. I leave as a suggestion for next SRs.

4 – Ok.

5 – Following the good practices of SRs, if the authors set explicitly the language as an exclusion criterion, this decision has to be justified. However, if the authors are unable to get the data from an article in an understandable language, these studies can be excluded during the screening or eligibility stages.  

6 to 11 – Ok.

12 – The removal of this study is not necessary since it fills the inclusion criteria. The suggestion made was only to address/discuss the fact that one study possesses almost 60% of the total amount of the participants. I do apologize if the comment was not clear.

13 and 14 – Ok. 

Author Response

2 – I apologize if the comment was not clear. For the reviewer’s point of view, the aims (line 119 – 122) are too broad for a systematic review. The authors should point out, as the aim of the study, the expected outcomes (e.g., what knowledge and precaution standards were researched), that is, knowledge of the environment, Gloves recommendation, knowledge of Precaution standards and knowledge of Invasive procedures.

Research reports related to knowledge and precaution standards among MIPs were compiled and interpreted in this study.

5 – Following the good practices of SRs, if the authors set explicitly the language as an exclusion criterion, this decision has to be justified. However, if the authors are unable to get the data from an article in an understandable language, these studies can be excluded during the screening or eligibility stages.

During screening or eligibility, studies that do not provide data in an understandable format can be excluded.